

# MARES Project: Hydrographic data of the San Jorge Gulf from R/V *Coriolis II* cruise in 2014.

Juan Cruz Carbajal[1], Marcela Charo[2], Andrés Luján Rivas[1], and Cèdric Chavanne[3]

[1]Centro para el Estudio de Sistemas Marinos, CONICET, Chubut, Argentina
[2]Departamento Oceanografía, Servicio Hidrografía Naval, Buenos Aires, Argentina
[3]Institut des sciences de la mer de Rimouski, Université du Québec à Rimouski, Canada

*Correspondence to:* Juan Cruz Carbajal (carbajaljuancruz@gmail.com)

**Abstract.** PROMESse (Multidisciplinary program for the study of the ecosystem and marine geology of San Jorge Gulf and the coast of the Province of Chubut) was an international cooperation research program among the Ministry of Science and Technology (MINCyT), the National Scientific and Technical Research Council (CONICET), the Province of Chubut (Argentina) and the University of Quebec at Rimouski (UQAR/ISMER, Canada). Within the framework of this program two projects were carried out, MARES (Marine Ecosystem Health of the San Jorge Gulf: Present status and Resilience capacity) and MARGES (Marine Geology). The main goal of MARES was to conduct a comprehensive study of the dynamics of physical, chemical and biological parameters vitals for the San Jorge Gulf ecosystem. The observational component of this project consisted on a multidisciplinary oceanographic cruise on board of the research vessel *Coriolis II* in Feb. 2014 integrated by three legs designed to identify and characterize areas of high primary productivity, which will serve as indicators of the ecosystem health. This paper reports the hydrographic data collected during the second leg of the *Coriolis II* cruise. This leg was aimed to study the frontal dynamics associated to a region of high tidal dissipation rate south of the Gulf, and to study the vertical displacements of the pycnocline at a fixed site in the center of the Gulf mouth. To this end, high-resolution data was collected in the southern tidal front, including quasi-continuous CTD vertical profiles, underway surface temperature and salinity, Scanfish II CTD and shipboard ADCP data. The data sets are available in the National Oceanographic Data Center (NODC) from NOAA. DOI: https://doi.org/10.7289/V5MP51J2

## 1 Introduction

The San Jorge Gulf (SJG) is a semi-open basin of approximately 40000 $km^2$ located between 45° S and 47° S with depths slightly over than 100 m in the central region (Fig. 1). Its broad mouth extends 230 km from Bahía San Gregorio to Cabo Tres Puntas along the meridian 65° 45' W, connecting the Gulf with the Argentine Continental Shelf through a sill that increases in the S-N direction, reaching a maximum depth of $\sim$ 60 m near 46° 48' S (Fig. 1). This geomorphological feature in interaction with tidal mixing produces changes of well-stratified conditions to well-mixed conditions within a few kilometers during the warm seasons.

The SJG circulation is driven by intense westerly winds and high amplitude tides (Palma et al., 2004; Tonini et al., 2006; Moreira et al., 2011). Estimates of tidal energy dissipation by bottom friction derived from numerical models results (Glorioso



and Flather, 1995, 1997; Palma et al., 2004; Moreira et al., 2011) suggest that most of the dissipation occurs at the mouth of the SJG, mainly in the southeast region. The dissipation rate is high enough to break up the seasonal thermocline and give rise to the formation of an intense tidal front. Due to its configuration and variability, the tidal front enhances the biological productivity nearby (Glembocki et al., 2015), plays a key role in the development of ecological processes and is closely related to fishery

resources (Acha et al., 2015; Alemany et al., 2014). Studying the frontal variability, both spatial and temporal, is essential to understand the mechanism responsible for that enhancement and to define main frontal properties related to biological effects. Thus, the use of a high-resolution sampler system was key to evidence the high-frequency frontal variability (Carbajal et al., in review).

The main objective of MARES leg 2 was to evaluate the high-frequency variations of the Southern Tidal Front (STF) of

the SJG. In order to achieve this purpose, a high-resolution sampling was carried out for a complete tidal cycle during three tidal states (see Sect. 2.1). Knowledge of mesoscale variability is not only crucial to interpreting the biological influence of the fronts (Landeira et al., 2014), but it will also contribute to the establishment of new conservation strategies and the management of marine resources. In this article, we will describe the cruise design and the procedures used for the acquisition, calibration and the processing of the dataset obtained during MARES leg 2.

## 15  2   Field measurements and equipment

Two types of surveys were carried out between Feb. 4-10 2014 on board of the Canadian research vessel (R/V) *Coriolis II* during MARES leg 2 cruise: one located in the STF region and another in a fixed position near the center of the Gulf mouth. While towed undulating vehicle systems have been used by investigators in the field (Twardowski et al., 2005; Brown et al., 1996), the PROMESse program was the ideal framework for the application of new technologies in the Argentine Continental

Shelf such as a towed undulating vehicle, achieving unprecedented high-temporal and spacial-resolution data in the region, particularly in the frontal zone. Table 1 summarizes the characteristics of the sensors used in each instrument and which are described in the following sections. Date and time from data sets are reported in Coordinated Universal Time (UTC).

### 2.1   Sourthern Tidal Front observations

Eighteen cross-front transects (six in late spring tide (Feb. 5), six in intermediate tide (Feb. 8-9) and six in early neap tide

(Feb. 9-10) were occupied in the STF using a towed undulating vehicle EIVA Marine Survey Solutions model Scanfish II (http://aquaticcommons.org/3106/1/ACT_WR07-01_Tower_Vehicles.pdf), fitted with a modular CTD Sea-Bird Electronic (SBE) model 49FastCAT (16 Hz) and a combined fluorometer and turbidity sensor WetLabs model ECO FLNTU (8 Hz). The modular CTD has no memory nor internal batteries, and does not support auxiliary sensor inputs either. Therefore, the ECO FLNTU could not be directly associated with the CTD data and thus the fluorescence and turbidity signal was acquired without the

corresponding date, time and position. Nevertheless, it was possible to link both signals to get the missing data in a post-processing. The sections length ranged between 17.4 km to 63.1 km, which is equivalent to 1:09 h and 4:33 h of transit (see Table 2 for details). The sections occupied in late spring and early neap tide covered an area of approximately 29.8 km (NW-



SE) by 15.1 km (NE-SW) during a semi-diurnal tidal cycle each (Figs. 1c and 1e, respectively), while the intermediate tide survey consisted of a single transect (T1) occupied six times, back and forth, also during a semi-diurnal tidal cycle (Fig. 1d). Surveys detail above are shown in Fig. 1. The horizontal separation of Scanfish sawtooth profiles was approximately 81 m–291 m, the latter largely dependent on bottom depth and the condition in the sea surface, descending (ascending) the vehicle at an

absolute rate of nearly 0.9 ms$^{-1}$.

On board, the towed vehicle was monitored through the roll and pitch sensors. The vehicle attitude was governed through two rear-mounted flaps and depths were provided by the CTD pressure sensor (Brown et al., 1996). The data collected with the Scanfish II provided a quasi-synoptic spatial and temporal resolution to characterize the influence of the high/low tide behavior and determine the front displacements relative to the phase of the tide.

## 2.2 Fix Station observations

From Feb. 6 2014 17:04 h UTC to Feb. 8 2014 04:01 h UTC, a time series was carried out in a fixed station (FS) near to the center of the SJG mouth (45° 56' S, 65° 33' W). The time series consisted of thirteen quasi-continuous full depth CTD-rosette casts collected approximately every 2:55 h during 34:57 h using a CTD Sea-Bird model SBE911*plus*, along with the plankton net and video plankton recorder (VPR). In addition, two sequential sediment traps were moored above and below the

pycnocline during seven days (Feb. 7-13). The data sets of sediment traps, plankton nets, and VPR are not reported in this data collection. The objective of this survey was to monitor the pycnocline displacements in the water column and to determine the mechanisms responsible for these vertical movements.

## 2.3 Complementary observations

An underway CTD Sea-Bird model SBE19*plus* (4 Hz) coupled with a Seapoint fluorescence sensor was used to identify the

position and orientation of the STF and remained operational throughout the entire cruise. Underway data were recorded every 10 s along the tracks.

Direct velocity measurements were collected with a Teledyne RD Instruments (TRDI) 150 kHz Ocean Surveyor hull-mounted Acoustic Doppler Current Profiler (ADCP). The seabed map and the distribution of biological species in the water column were achieved using a hull-mounted scientific echo-sounder SIMRAD model EK60 working in multiple frequencies

(38 kHz, 120 kHz and 200 kHz).

## 3 Hydrographic data

### 3.1 CTD Vertical Profiles

An additional cross-frontal transect was occupied across the STF (on Feb. 5 at night and Feb. 8 at late afternoon) to study the biological and chemical characteristics of the water column. Each realization consisted of five CTD vertical profiles spaced at

distance intervals of ∼ 4.9 km, using a CTD Sea-Bird model SBE911*plus*, equipped with oxygen, pH, fluorescence, nutrients,



photosynthetically active radiation (PAR), beam transmission, altimeter and $pCO_2$ sensors (Table 1). The altimeter sensor was used to determine distance to the bottom. Most vertical profiles reached to within $\sim 9$ m off the bottom. Oxygen, pH and fluorescence sensors were calibrated as described in Sect. 3.3. Data from the remaining sensors are reported based on factory calibrations only. Particularly, the $pCO_2$ sensor did not work properly (it recorded a constant value of $-0.00088$).

Down and up-casts profiles are reported in this dataset, but it has to be mentioned that during downcast the CTD sensors measure the water column with minimal interference from the underwater package.

## 3.2  Water Samples

A Carousel Sea-Bird model SBE32 rosette package with twelve 12 L Niskin bottles was employed during the cruise. At pre-defined depths, water samples were collected for the determination of salinity, dissolved oxygen (DO), nutrients, pH and

chlorophyll-a (Chl) concentrations in the water column. In addition, they were used to calibrate the ancillary sensors of the CTD, as showed in Sect. 3.3.

Due to the lack of a salinometer on board, salinity water samples were drawn in small glass flasks and sealed with insulation tape. Fourteen days after the cruise ended, salinity samples were determined at the Instituto Nacional de Investigación y Desarrollo Pesquero (INIDEP) with a Guildline Autosal 8400B salinometer based on the technical specifications of the manual

(Guildline Instruments, 2004). All salinity samples showed a positive bias ($0.009\pm0.007$ psu, $N = 35$) when compared to CTD salinity values. The water samples most likely experienced evaporation during this period and therefore the salinity measurements were overestimated. Bottle salinity data were considered questionable, and because the CTD conductivity sensor was factory calibrated in 2013, the bottle salinity data were not used to calibrate it. Salinity values were calculated and reported in Practical Salinity Units (UNESCO IWG, 1981).

DO water samples were drawn in DBO glass bottles with frosted neck to avoid evaporation. DO concentrations were determined with a modified Winkler method (Carpenter, 1965) using an automatic Metrohm volumetric titrator.

Nutrients water samples were drawn in 250 mL plastic bottles and frozen immediately at -20° C without previous filtration, until they were analyzed at the Laboratorio de Oceanografía Química y Contaminación de Aguas (LOQyCA) of the Centro Nacional Patagónico (CENPAT). Analysis of four macro-nutrients (nitrate, nitrite, silicate and phosphate) concentrations were

determined using colorimetric techniques on Skalar San Plus autoanalyser (Skalar Analytical® V.B, 2005a, b, c), according to the methods described in Strickland and Parsons (1972).

Chl concentrations were determined fluorometrically by filtering 500 to 1500 mL seawater samples through 25 mm Whatman GF/F (0.7 µm porosity) glass fiber filters. Chl and phaeopigments were extracted on board in 90 % acetone for a period of 12 to 24 h in the dark. Fluorescence measurements were performed on board in the dark with a Turner Designs model 10-AU

fluorometer according to the Parsons et al. (1984) method.

pH of water samples were measured on board using a YSI 556 MSI multiparameter probe.




### 3.3 CTD ancillary sensors calibration

The SBE43 oxygen sensor has a very stable electronic system, therefore, any loss of its accuracy with time is primarily attributed to fouling of the membrane, either biological or waterborne contaminants (e.g., oil). The manufacturer provides an algorithm to adjust the drift by using a quality reference sample such as Winkler titrated and the SBE43 measured DO
concentrations at the times the water samples were collected (Sea-Bird Electronics Inc, 2012). DO differences larger than 1 $mLL^{-1}$ were discarded and not used in the DO sensor calibration. The standard deviation of the residuals after calibration was approximately 0.08 $mLL^{-1}$ ($N = 48$). Figure 2 presents the residual (CTD DO - Winkler DO) in $mLL^{-1}$ before and after calibration.

The MBARI-ISUS sensor is designed mainly to determine nitrate concentrations in the 200-400 nm range of the ultraviolet-
spectrum, as the main nutrients required for growth of phytoplankton. A post-cruise inspection of the vertical distribution of nitrate together with the nutrients water samples were carried out in the frontal zone and in the FS. The analysis suggested that the nutrient sensor did not work properly, showing negative values of nitrates in some profiles. Thus, the data of this sensor are reported based on its factory calibration.

The fluorescence sensor calibration was performed using a linear adjustment with Chl concentration from water samples to
obtain the calibration coefficients. In the same way, pH water samples were fitted to pH sensor data. The squared correlation coefficient was $R^2 = 0.891$ (N=37) and $R^2 = 0.908$ (N=54), respectively.

### 4 Scanfish II data calibration

In April 2015, the SBE49FastCAT sensor was sent to SBE factory for the post-cruise calibration. Thus, the pre- and post-cruise calibration was used to generate a *slope* (in conductivity) and an *offset* (in temperature) corrections for these data, following
Sea-Bird Electronics Inc (2010). The laboratory calibration showed drift corrections of -0.00010 psu month$^{-1}$ and -0.00017 °C year$^{-1}$; corresponding to a conductivity *slope* value of 1.00010 and a temperature *offset* value of 0.00052 °C, respectively. Fluorescence and turbidity data from the ECO FLNTU sensor are reported based on factory calibrations only.

### 5 Scanfish II and CTD data processing

Scanfish and CTD data were post-processed using SBE Data Processing software routines (v. 7.23.2, Seasoft-Win32, http:
//www.seabird.com/software/software). The processing sequence for the SBE911*plus* did not follow the *typical* sequences suggested by the manufacturer because the 'scans to average' parameter was set to 24 in the configuration file used on board and so the raw vertical profiles were stored at 1 Hz averages. SBE technical support suggested skipping any filter steps, since the data was already averaged (see SBE manual for details, http://www.seabird.com/sites/default/files/documents/SBEDataProcessing_ 7.26.4.pdf). The Deck Unit was programmed to advance conductivity 0.073 s relative to pressure. Because oxygen data is also
systematically delayed with respect to pressure, several tests were carried out to determine the best aligment, which was set to



+3 s. Each CTD profile was then inspected and density inversions were removed and filled in by a linear interpolation of the original temperature and conductivity data. Finally, all derived parameters were recalculated at the interpolated pressure.

The processing sequence for the SBE49FastCAT was based on the manufacturer suggestions. Raw temperature and conductivity data are often misaligned with respect to pressure in areas with strong vertical stratification due to vertical temperature

gradient, causing spikes in derived variables, mainly in salinity. This misalignment, which depends on temperature, conductivity and pressure, was partially corrected with the SBE Data Processing software *align* routine by advancing temperature +0.063 s relative to pressure. Then, a low-pass filter was applied to pressure, conductivity and temperature to smooth high frequency data. The main issue was the modeling of the thermal inertia effects within the conductivity cell (Lueck, 1990; Lueck and Picklo, 1990), which induces changes in the derived salinity values. These thermal effects are contemplated in the algorith-

mic *cell-thermal mass* of the SBE Data Processing software, through the coefficients $\alpha$ (initial magnitude of the fluid thermal anomaly) and $\beta^{-1}$ (relaxation time of the fluid thermal anomaly). The salinity signal after applying both the standard SBE coefficients ($\alpha = 0.03$ and $\beta^{-1} = 7$) and those proposed by Lueck and Picklo (1990) ($\alpha = 0.028$ and $\beta^{-1} = 9$) was very similar. The highest difference between the salinity profiles before and after applying the *cell-thermal mass* algorithm (0.03 psu) was found at the depth of the pycnocline (z$\sim$ 38 m). Modeling the anomaly was particularly challenging for the sawtooth profiles

considering the results from Lueck and Picklo (1990), who found that the anomaly persists 45 s after crossing the thermocline. Following the manufacturer suggestions, it was decided not to apply the *cell-thermal mass* algorithm to the conductivity signal at all. Finally, the derived variables were calculated. Remaining salinity spikes could be related to shed wakes from the CTD package that mix up the surrounding water. Different experiments with the *median filter* routine on derived variables data were performed to minimize spiking. A window size of $3 \cdot 16 \mathrm{Hz} = 48$ scans was used and the filter was applied consecutively three

times for conductivity, temperature, salinity and density data.

## 6  Underway data

A linear least squares fit was made for each variable between the CTD profiles data extracted at the 2 dbar level during down- and up-cast and the underway data to determinate an *offset* and a *slope* for the underway data corrections. This calibration was conducted using only CTD and underway data collected simultaneously in time and space (35 data points in total). Figure 3

shows the pre- and post-calibrated differences between CTD and underway data for each variable.

Commonly, spurious data can be recorded by pump malfunction that alters the flow of water through the internal conduits or by chemical or biological depositions in the measurement cells or filters. An inspection of the data was held and questionable data were rejected. Finally, to smooth the noise in the underway calibrated data caused by flow rate disturbances, temperature and conductivity were filtered by applying the *median filter* routine from the SBE Data Processing software, using a window

size of 512 scans. This filter was carried out twice consecutively before the *derive* routine was applied.



## 7 ADCP data

The hull-mounted ADCP data was collected with the TRDI VMDAS software version 1.3. At the beginning of the cruise the vessel position was provided by a GPS Furuno GP-31, but no heading signal input was set. The problem was discovered on Feb. 6 at 0 h UTC after the late spring survey across the STF. Therefore, only along-track velocities can be used from this data.

The ancillary navigation input was then modified and GPS directional Applanix POS MV data was then set as input for the heading and attitude data.

The transducer depth and the blanking interval were 3.93 m and 4 m, respectively. The ADCP data were set to profile with a vertical bin size of 4 m (50 bins total) with single-ping bottom track enabled to a maximum depth of 500 m, and a ping interval of 2 s between ensembles. The data were then averaged to 3 min temporal resolution (approximately 0.6 km in average).

The ADCP data was processed using CODAS (Common Oceanographic Data Access System) software system v.3.1 developed at the University of Hawaii (Firing, 1995), https://currents.soest.hawaii.edu/docs/adcp_doc/codas_doc/.

## 8 Lessons learned

Prepare a detailed list including all measurements to be made, along with methodology, sampling and analytical instrumentation (Pollard et al., 2011). Collection of ocean data of the highest possible quality requires careful Quality Control and Quality

Assurance procedures for the underwater unit and all sensor components (https://www.oceanbestpractices.net/handle/11329/336). These control tasks should be carried out and if possible formally reported before the cruise starts.

The overall quality of CTD data collection depends on a number of factors, such as: sensor calibration; equipment performance; software configuration and bugs during acquisition and processing; hardware problem; etc.

The selection of instruments and sensors used for data acquisition was carried out before the cruise in the Québec – Ocean

Laboratory (UQAR/ISMER, Canada). They checked sensors calibration and equipment performance to acquire the best possible hydrographic data. However, some system presented problems during the cruise as were documented in this paper. The control tasks will be more carefully checked in the next cruises.

## 9 Data availability

The final dataset concatenates the different collections during the cruise, which are quasi-continuous CTD vertical profiles, the

underway surface temperature, salinity and fluorescence data, the Scanfish II CTD and FLNTU data, and the shipboard ADCP data. CTD vertical profiles and underway data are reported in standard Sea-Bird Data File (cnv) format. Converted files consist of a descriptive header followed by the calibrated-processed data in columns. Scanfish II CTD and FLNTU data are reported in comma-separated values (csv) format. One file for section and for sensor is provided for each survey across the STF (e.g., file named T1_lst correspond to CTD data for transect 1 in late spring tide and T1_FLNTU_lst correspond to FLNTU data

for transect 1 in late spring tide). Shipboard ADCP data also are reported in comma-separated values (csv) format, each file





name according to the CODAS output. The data are currently available at the US National Oceanographic Data Center, NOAA, under https://doi.org/10.7289/V5MP51J2.

**Appendix A: Scanfish II data gridding**

Since the data collected from the Scanfish have non-uniform spacing in both horizontal and vertical, the calibrated and post-
processed data of each section was transformed into a rectangular grid with a resolution of $593 - 610$ m in the horizontal and $0.6$ m in the vertical for a better post-analysis. After testing several geometries, the final grid geometry followed two steps. The vertical limits were set to the minimum and maximum depth reached by the Scanfish, corresponding to $0.063$ m and $85.131$ m, respectively. Then, as the length of each section was different, for each of them, the horizontal limits begins and ends where a 'V-shaped' profile begins or ends. A 'V-shaped' profile is the sum of a consecutive down and up-cast of the Scanfish. Kriging
method type point was used to interpolate the node values, with horizontal and vertical radius of $1.8$ km and $1.2$ m, respectively. Figure A1 focuses on one of the sections (T1) for late spring tide to illustrate the main features of the water column structure near the southern mouth of SJG, representative of average summer conditions.

*Competing interests.* The authors declare that they have no conflict of interest.

*Acknowledgements.* This program was financed by the Institut des sciences de la mer de Rimouski/Université du Québec à Rimouski (IS-
MER/UQAR) from Canada, the Consejo Nacional de Investigaciones Científicas y Técnicas (CONICET) and the Ministerio de Ciencia y Tecnología (MINCyT) from Argentina. We thank the Instituto Nacional de Investigación y Desarrollo Pesquero (INIDEP, Argentina) that made available the Autosal salinometer and the Laboratorio de Oceanografía Química y Contaminación de Aguas (LOQyCA, Argentina) for allowing the use of the water samples measurements of nutrients, pH and DO. We also acknowledge Valérie Massé-Beaulne's detailed expla-
nation of the measurement of Chl concentrations. We thank the crew and scientific staff of R/V *Coriolis II*. Finally, we wish to acknowledge
D. Valla for his useful comments and language editing.





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



**Table 1.** Summary of the sensors used on board R/V *Coriolis II* in MARES leg 2 (Feb. 4-10 2014).

| Instrument | Sensor | Model | Serial # | Calibration date |
|---|---|---|---|---|
| SBE911*plus* (CTD profile) | Temperature | SBE3*plus* | 5769 | Oct.22 2013 |
| | Pressure | Digiquartz with TC | 1168 | Nov.19 2013 |
| | Conductivity | SBE4 | 4244 | Nov.06 2013 |
| | Oxygen | SBE43 | 2766 | Nov.15 2013 |
| | pH | SBE18 | 1078 | Nov.20 2013 |
| | Fluorescence | ECO FL WetLabs | FLRT-3363 | Nov.11 2013 |
| | Nutrients | Satlantic MBARI-ISUS | 0184 | May.25 2013 |
| | PAR | Biospherical/Licor | 70455 | Nov.04 2013 |
| | Beam transmission | WetLabs C-Star | CST-1628PR | Jun.11 2013 |
| | $pCO_2$ | CONTROS | CO2-0610-001 | Apr.28 2011 |
| | Altimeter | PSA-916 | 61114 | - |
| SBE49FastCAT (Scanfish II) | Temperature | | 0226 | Jan.17 2011 |
| | Pressure | Strain gauge | 0226 | Jan.14 2011 |
| | Conductivity | | 0226 | Jan.17 2011 |
| | Fluorescence/Turbidity | ECO FLNTU WetLabs | FLNTURT-2037 | Oct.07 2010 |
| SBE19*plus* (Underway) | Temperature | | 4975 | Mar.07 2013 |
| | Pressure | Strain gauge | 4975 | Feb.28 2013 |
| | Conductivity | | 4975 | Mar.01 2013 |
| | Fluorescence | Seapoint Chlorophyll Fluorometer | 2803 | Apr.28 2006 |

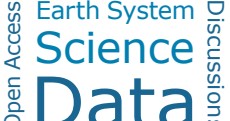

**Table 2.** Field measurements across the STF using the towed undulating vehicle Scanfish II. Arrows indicate the cruise path parallel to each transect ($\Rightarrow$: offshore the Gulf, $\Leftarrow$: into the Gulf). $\Delta x$ represents the transect length.

| | Transect | Path | Begin | | End | | Depth range (m) | $\Delta x$ (km) | # Scans |
| --- | --- | --- | --- | --- | --- | --- | --- | --- | --- |
| | | | Date (hh:mm) | lat./lon. (°) | Date (hh:mm) | lat./lon. (°) | | | |
| late spring tide | T1 | $\Rightarrow$ | Feb.5 (07:38) | -46.51,-65.97 | (10:01) | -46.61,-65.71 | 0.83-79.87 | 24.0 | 114505 |
| | T2 | $\Leftarrow$ | (10:13) | -46.59,-65.70 | (12:10) | -46.46,-66.01 | 2.40-81.36 | 28.8 | 111801 |
| | T3 | $\Rightarrow$ | (12:21) | -46.44,-66.00 | (13:55) | -46.54,-65.74 | 2.73-79.95 | 22.4 | 90001 |
| | T4 | $\Leftarrow$ | (14:08) | -46.51,-65.73 | (15:48) | -46.42,-65.96 | 6.11-79.96 | 21.2 | 96002 |
| | T5 | $\Rightarrow$ | (15:59) | -46.40,-65.94 | (17:43) | -46.51,-65.65 | 6.81-80.24 | 25.4 | 99601 |
| | T6 | $\Leftarrow$ | (17:59) | -46.49,-65.64 | (19:23) | -46.40,-65.86 | 4.17-79.74 | 20.4 | 81201 |
| intermediate tide | T1-1 | $\Rightarrow$ | Feb.8 (23:58) | -46.50,-66.00 | Feb.9 (01:32) | -46.60,-65.76 | 0.50-82.97 | 21.8 | 90003 |
| | T1-2 | $\Leftarrow$ | (01:46) | -46.60,-65.76 | (02:55) | -46.52,-65.95 | 1.54-84.37 | 17.4 | 66001 |
| | T1-3 | $\Rightarrow$ | (03:05) | -46.52,-65.97 | (04:37) | -46.61,-65.72 | 1.30-82.71 | 22.1 | 88001 |
| | T1-4 | $\Leftarrow$ | (04:45) | -46.61,-65.71 | (06:46) | -46.50,-66.01 | 2.66-80.68 | 27.3 | 116001 |
| | T1-5 | $\Rightarrow$ | (07:07) | -46.50,-66.00 | (08:45) | -46.61,-65.72 | 2.74-80.44 | 25.5 | 94002 |
| | T1-6 | $\Leftarrow$ | (09:00) | -46.62,-65.70 | (11:19) | -46.49,-66.04 | 2.04-85.13 | 30.4 | 134002 |
| early neap tide | T1 | $\Rightarrow$ | (11:34) | -46.49,-66.04 | (13:48) | -46.61,-65.71 | 4.36-80.24 | 29.7 | 128801 |
| | T2 | $\Leftarrow$ | (14:02) | -46.59,-65.70 | (16:00) | -46.46,-66.01 | 2.37-80.38 | 28.6 | 114001 |
| | T3 | $\Rightarrow$ | (16:13) | -46.44,-66.00 | (18:06) | -46.56,-65.68 | 2.65-80.34 | 28.3 | 108801 |
| | T4 | $\Leftarrow$ | (18:20) | -46.54,-65.66 | (21:02) | -46.39,-66.03 | 2.49-81.49 | 34.0 | 156001 |
| | T5 | $\Rightarrow$ | (21:17) | -46.37,-66.01 | (23:26) | -46.51,-65.65 | 0.06-80.27 | 32.0 | 124002 |
| | T6 | $\Leftarrow$ | (23:40) | -46.49,-65.63 | Feb.10 (04:13) | -46.34,-66.00 | 1.71-83.93 | 63.1 | 262406 |



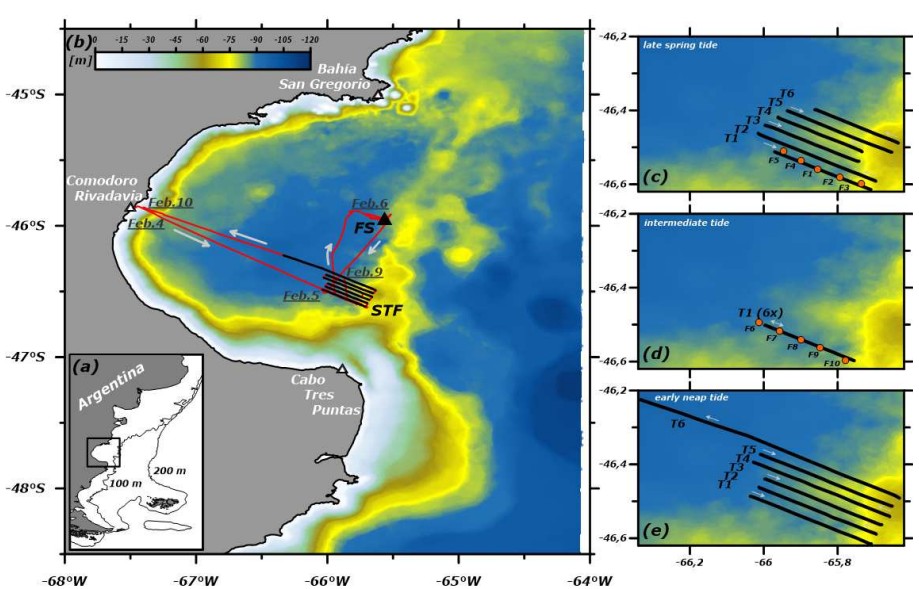

**Figure 1.** Study area (a) Patagonia shelf; (b) San Jorge Gulf, the underway track for MARES leg 2 (red line), the Scanfish transects across the STF (black lines) and the FS (black triangle). A zoom of the survey across the STF for (c) late spring tide, (d) intermediate tide, showing the CTD vertical profiles (orange circles); and (e) for early neap tide. Bathymetry is shown as shaded colors, highlighting the bank south of the Gulf, where depths range from 45 to 75 m. Arrows in light gray indicate the cruise path, particularly in d) the double arrow references the survey back and forth over T1. STF: Southern Tidal Front. FS: Fixed Station.



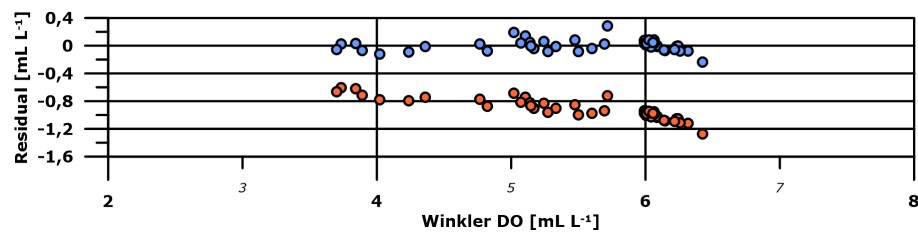

**Figure 2.** Distribution of dissolved oxygen residuals versus dissolved oxygen concentration (both in $\mathrm{mLL^{-1}}$), before (orange dots) and after (blue dots) the SBE43 sensor calibration.





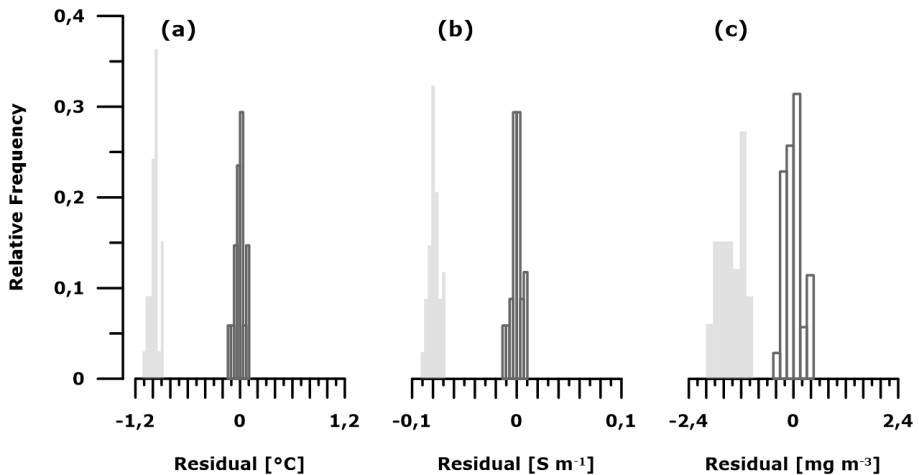

**Figure 3.** Relative frequency of (a) temperature, (b) conductivity and (c) fluorescence residuals before (gray shaded bars) and after (black solid bars) the underway calibrations.




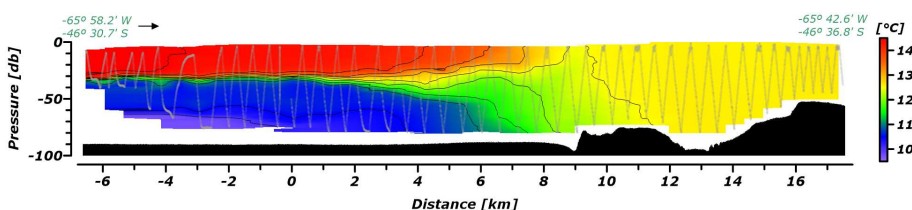

**Figure A1.** Vertical section of temperature in late spring tide for transect 1. The horizontal scale represents distance in km from an arbitrary zero position, whereas the profile of the seabed is derived from the ship echo-sounder EK60. The consecutive 'V-shaped' profiles of the Scansifh II are marked in gray, with a dot every 10 data points, and the cruise path with a black arrow.