# Peer review of "MARES Project: Hydrographic data of the San Jorge Gulf from R/V *Coriolis II* cruise in 2014."

_Earth System Science Data, 2018_

## Referee Comment (RC1) · Anonymous Referee #1 · 15 Aug 2018

Journal: Earth System Science Data (ESSD) Title: MARES Project: Hydrographic data of the San Jorge Gulf from R/V Coriolis II cruise in 2014 Author(s): Juan Cruz Carbajal, Marcela Charo, Andrés Luján Rivas, and Cèdric Chavanne MS No.: essd-2018-75 MS Type: Peer-reviewed comment

General Comments This manuscript reports, the hydrographic data collected during the second section of an oceanographic cruise achieved at the Patagonian coast, in the San Jorge G in the frame of program called MARES. The objective of the program is to study the frontal dynamics associated to a region of high tidal dissipation rate south of the Gulf, and analyze the vertical displacements of the pycnocline at a fixed site in

the center of the Gulf mouth. High-resolution data was collected in the southern tidal front, including quasi-continuous CTD vertical profiles, underway surface temperature and salinity, Scanfish II CTD area used for data acquisition.

Some data were not properly calibrated (nutrients, salinity) because of problems in sampling or control failures. As the authors say "The control tasks will be more carefully checked in the next cruises".

The organization of the MS is good.I only suggest to change a paragraph from section2.2 to section 2.3- I suggest to make some comparison between historical data nad observations of this cruise, specially referred to nutrients and chlorophyll data. It should be useful to compare Temperature vs Nitrates from historical data (if available) This MS would be acceptable for publication ESSD after minor changes Following I make some editorial comments.

Editorial comments Abstract Line 6. I suggest drive instead of conduct L 9. Isuggets to change leg (s) bye transect in all the MS

Section 2 L 17. Supress in the field L 20. Supress "towed undulating vehicle" by this vehicle

Section 2.2 L14 to16. I suggest to change the sentence:" along with the...collection" to 2.3 (Complementary observations section)

Secion 3-3 L. 6. In the international system of units, units for DO should be $\mu$mol/kg. L. 16.. Specify standard errors

---

## Author Comment (AC1) · 29 Aug 2018

We thank Reviewer #1 for his/her comments. Below we respond to each of the reviewer comments.

With regard to the comparison between historical data and observations of this cruise, we have included a paragraph in Sect. 3.2 line 33.

'Nutrients, DO, chl and pH data from the water samples collected in MARES leg 2 cruise were compared with historical data available in the Argentine Oceanographic Data Center (CEADO, http://www.hidro.gov.ar/ceado/ceado.asp) for austral summer

(Jan-Feb-Mar) in the region of interest. CEADO archives data originated by national and international research institutions. Water samples observations were consistent with historical observations.'

Abstract Line 6. Done. L 9. The word 'leg' refers to the different stages of the MARES project. We believe it is the word to name the oceanographic cruise. 'Transects' is used for oceanographic sections.

Sect. 2 L. 17 and L. 20. Done.

Sect. 2.2 L 14 to 16. Done.

Sect. 3.3 L. 6. The Sea-Bird application note for the DO calibration uses the ml l-1 units. However, we have included in Section 3.3 line 9 the conversion unit to $\mu$mol/kg obtained from SBEData processing manual (https://www.seabird.com/). L. 16. We have specified the standard deviations post-calibration for the fluorescence and pH sensors of the CTD.

In addition, we have detected an error in line 7 from Sect. 3.3. We have changed 'DO differences larger than 1 ml l-1 were discarded and not used in the DO sensor calibration.' for 'DO differences greater than one standard deviation were discarded and not used in the DO sensor calibration.'

Please also note the supplement to this comment:
https://www.earth-syst-sci-data-discuss.net/essd-2018-75/essd-2018-75-AC1-supplement.pdf
* * *
[Figure]

**Supplement:**

[revised manuscript text omitted]

Nutrients, DO, chl and pH data from the water samples collected in MARES leg 2 cruise were compared with historical data available in the Argentine Oceanographic Data Center (CEADO, http://www.hidro.gov.ar/ceado/ceado.asp) for austral summer

(Jan-Feb-Mar) in the region of interest. CEADO archives data originated by national and international research institutions. Water samples observations were consistent with historical observations.

**3.3 CTD ancillary sensors calibration**

The SBE43 oxygen sensor has a very stable electronic system, therefore, any loss of its accuracy with time is primarily attributed to fouling of the membrane, either biological or waterborne contaminants (e.g., oil). The manufacturer provides an algorithm to adjust the drift by using a quality reference sample such as Winkler titrated and the SBE43 measured DO concentrations at the times the water samples were collected (Sea-Bird Electronics Inc, 2012). DO differences greater than one standard deviation were discarded and not used in the DO sensor calibration. The standard deviation of the residuals after calibration was approximately $0.08 \, \mathrm{ml \, l}^{-1}$ ($N = 48, DO(\mu\mathrm{mol/kg}) = 44660 \cdot DO(\mathrm{ml/l})/(\sigma_{theta} + 1000)$). 
[revised manuscript text omitted]

---

## Referee Comment (RC2) · Anonymous Referee #2 · 10 Dec 2018

Dear Authors, please here my review of a manuscript entitled "MARES Project: Hydrographic data of the San Jorge Gulf from R/V Coriolis II cruise in 2014". This manuscript presents data from a 7-day hydrographic leg in the Gulf of San Jorge, Argentina, during 2014. The manuscript is organized as if 2 types of "independent" data are presented: 1) a fixed hydrographic station in the center of the Gulf and 2) data from an undulating CTD data across a frontal area. These data are complemented by data from an underway CTD and vessel-mounted ADCP. Mid-way in the reading, we however discover that a third data set corresponding to CTD cast in the southern frontal is also provided. Data from water samples (Niskin bottles), Simrad echosounder (3 frequencies), sediment traps, VPR, plankton nets, etc. are also discussed/mentioned but not presented.

While there is no doubt that the cleaning and processing of these data represent a lot of work in term of methodology, including the cross-calibration of instruments, I am afraid that the nature of the data is of little significance for a journal such as ESSD. Therefore I cannot recommend the publication. Explanations are provided in the following.

GENERAL COMMENTS

1. My main concern is the very little originality, significance usefulness of these data, which are important criteria for ESSD. The data are from a rather short study (a week) in a localized area of the globe. Just think for example of the 14 CTD casts at the fixed station and 9 CTD casts at the Southern front area... these are little numbers that have a very limited usage... Overall, I find that the authors were not successful in promoting their dataset as a useful one for future users around the globe. This manuscript is closer to a mission report or a methodology section of a paper rather than an original piece of work. For example, I would imagine that the submitted paper referred to in the study (Carbajal et al, submitted to The Oceanography Society), would be a natural place to find most of the information provided here.

2. My second concern is the lack of integration of the data. The abstract mentions that these data are from a leg that aimed to study frontal dynamics, but this aspect (e.g., data useful to study fronts) is not promoted in the manuscript. There is also an evident disconnection between the data from the fixed station and those from the undulating CTD that targets the frontal area. This disconnection is not appealing for future users.

3. The lack of integration of the data is also clear in the way the data are presented as if they were just archived as they go out of the instrument (i.e., low hierarchical level). For example, the CTD casts are not binned on a regular vertical grid and the upcasts of the CTD not even removed (even though it is said that they are less reliable than the downcast!, see p.4 L5). The data have also been archived for each instrument separated without any attempt to merge them in higher hierarchical level files (for example in a netCDF file). This is below my expectations for a journal such as ESSD.

[Figure]

4. I also find the paper not well organized (this point is minor compared the previous ones). The Section 2 aims to introduce the field measurements and equipment. However, all kind of new instruments or data are also introduced in the following Sections 3 to 7. I also think Section 8 is of little use. It is also quite annoying that most of the data discussed are not even provided. For example, the largest section of the manuscript is "3.2 - Water samples.". But these data are not provided... More of these problems are provided as specific comments below.

SPECIFIC COMMENTS

- p.3 L.30: "it was possible to link both signals to get the missing data in a post-processing." How this is done?

- p.3 L.23: "The seabed map and the distribution of biological species in the water column were achieved using a hull-mounted scientific echo-sounder SIMRAD model EK60 working in multiple frequencies (38 kHz, 120 kHz and 200 kHz)." Why mentioning this if the data are not provided??

- p.3 L.28: "An additional cross-frontal transect was occupied across the STF (on Feb. 5 at night and Feb. 8 at late afternoon) to study the biological and chemical characteristics of the water column." This comes out of nowhere and was not introduced in the the section that aimed to present the field measurements.

- p.5 L.21: are the offset values provided smaller than the resolution of the instrument?

- Figures 2 and 3: All kind of data calibration are discussed in the manuscript. Why this choice of only presenting SBE43 and underway CTD? We don't learn much here.

---

## Author Comment (AC2) · 5 Feb 2019

Dear Anonymous Referee #2. We are very gratefully for your useful comments about our manuscript.

The initial organization of the manuscript was based on two different and close surveys, 1) a fixed hydrographic station and 2) a survey in the frontal zone, but it's true that a new dataset (CTD cast) is introduced as the paper unfolds. We followed your suggestion and we have made some modifications in the organization (see General Comment number 4). We agree that the cleaning and processing of the data demand a lot of work especially knowing that the cruise involved different working groups. We believe

that the data set from this hydrographic leg will contribute to understanding the natural ecosystem functioning of the San Jorge Gulf since there is a lack of information on the physical-biological coupling at high-frequency scales, key for ecological processes and fishery resources. Furthermore, as part of the Patagonian Shelf, the seasonal interaction between both gulf and inner shelf water masses enhances the productivity of one of the Large Marine Ecosystems of the globe (Sherman, K., & Adams, S. (Eds.). (2010). Sustainable development of the World's large marine ecosystems during climate change: a commemorative volume to advance sustainable development on the occasion of the presentation of the 2010 Göteborg Award. International Union for Conservation of Nature and Natural Resources). In the following, we provide the answers to all your comments.

GENERAL COMMENTS 1. The study was based on a program that began in Feb 2014 (as we have mentioned in the abstract) and then continued in the spring of 2016 and 2017 in the framework of a national project. In those two last surveys, hydrographic observations were collected in the frontal region but the towed undulating vehicle was not available, as we mentioned in Section 2, so the article focuses on the originality of the data set from the Scanfish II in a region with high biological impact. The methods and materials that support the provided data set are described in detail in the manuscript (standard SBE calibration/processing, CODAS processing system for ADCP and strict procedures for Water Samples analysis) so potential users have standard data quality ('state of the art') for future work. In addition, a complete metadata explaining the precision of each instrument and the data quality flags were presented at the NOAA. The work of this manuscript was designed to group the main dataset of the leg 2 and to describe in detail the procedures to reach a high-quality standard as the ESSD intends.

2. In the introduction, we have described the importance of studying the frontal variability and the potential impacts in the biological community (several papers are cited). At present, we are working with the data set of the fixed station and we assume that physical processes that occur in frontal region (as nutrients supply to the euphotic zone,

Carbajal et al., 2018) are also visible in the fixed station. Therefore, it is expected that the mechanisms could be similar to that observed in the southern tidal front, as we have mentioned in Section 2.2.

3. Regarding to this point, we have made some changes in the data of the CTD vertical profiles and the underway data format. The CTD vertical profiles data were averaged at 1 dbar pressure intervals and reported only the downcast and we have reprocessed the underway data, as we explained in Section 6. We are also going to upload this new version of the data set at NODC in CSV format.

4. We have taken your suggestion and have modified the structure of the manuscript. In Section 2 we finally present the field measurements and the instruments for the two surveys (tidal front and fixed station) and we mention the complementary observations (underway, ADCP) as a subsection. Regarding the water samples, we have provided meticulous detail since we used them for the calibration of the ancillary sensors of the CTD vertical profiles in order to reflect the 'state of the art' as part of the criteria quality of ESSD. Potential users of the dataset will know in detail how the ancillary sensors were calibrated. Section 8 was added as an initial suggestion from the Topical Editor of ESSD.

SPECIFIC COMMENTS - We included the following sentences (p.3 L.10) to explain the procedure: Nevertheless, it was possible to link both signals to get the missing data in a post-processing using complete data (date, time and position) of Scanfish CTD and the sampling intervals of the two sensors employed simultaneously to link the missing data to the corresponding record of the FLNTU data.

- We have deleted this sentence.

- No, the resolution of the SBE 49FastCAT temperature sensor is 0.0001 °C < 0.00052 °C (offset value correction).

- We have presented as examples the residual corrections before and after calibration

of SBE43 oxygen sensor and underway sensors in order to show how important it is to produce calibrated data.

Please also note the supplement to this comment:
https://www.earth-syst-sci-data-discuss.net/essd-2018-75/essd-2018-75-AC2-supplement.pdf

**Supplement:**

**MARES Project: Hydrographic data of the San Jorge Gulf from R/V *Coriolis II* cruise in 2014.**

Juan Cruz Carbajal[1], Marcela Charo[2], Andrés Luján Rivas[1], and Cédric Chavanne[3]

[1]Centro para el Estudio de Sistemas Marinos, CONICET, Chubut, Argentina
[2]Departamento Oceanografía, Servicio Hidrografía Naval, Buenos Aires, Argentina
[3]Institut des sciences de la mer de Rimouski, Université du Québec à Rimouski, Canada

*Correspondence to:* Juan Cruz Carbajal (carbajaljuancruz@gmail.com)

**Abstract.** PROMESse (Multidisciplinary program for the study of the ecosystem and marine geology of San Jorge Gulf and the coast of the Province of Chubut) was an international cooperation research program between the Ministry of Science and Technology (MINCyT), the National Scientific and Technical Research Council (CONICET), the Province of Chubut (Argentina) and the University of Québec at Rimouski (UQAR/ISMER, Canada). Within the framework of this program two projects were carried out, MARES (Marine Ecosystem Health of the San Jorge Gulf: Present status and Resilience capacity) and MARGES (Marine Geology). The main goal of MARES was to drive a comprehensive study of the dynamics of physical, chemical and biological parameters, which are vital for the San Jorge Gulf ecosystem. The observational component of this project consisted on a multidisciplinary oceanographic cruise on board of the research vessel *Coriolis II* in Feb. 2014 integrated by three legs designed to identify and characterize areas of high primary productivity that will serve as indicators of the ecosystem's health. This paper reports the hydrographic data collected during the second leg of the *Coriolis II* cruise. This leg's aim was to study the frontal dynamics associated to a region of high tidal dissipation rate south of the Gulf and to study the vertical displacements of the pycnocline at a fixed site in the center of the Gulf mouth. To this end, high-resolution data (Scanfish II) and quasi-continuous CTD vertical profiles were collected in the southern tidal front, and 'yo-yo' CTD casts were occupied at a fixed location. Moreover, complementary data from underway surface CTD and shipboard ADCP were collected during the cruise. The data sets are available in the National Oceanographic Data Center (NODC) from NOAA. DOI: https://doi.org/10.7289/V5MP51J2

**1   Introduction**

The Patagonian Shelf is ranked as the fifth most shallow sea region in the global distribution of tidal energy dissipation, accounting nearly 112 GW for the M2 tidal component and is known for its large tidal amplitudes and the speed of the tidal wave (Egbert and Ray, 2001; Simpson and Bowers, 1981; Glorioso and Flather, 1995, 1997; Miller, 1966; Cartwright and Ray, 1991; Webb, 1973; Forbes and Garraffo, 1988; Rivas, 1994). It is also the host of extremely rich persistent-and-seasonal frontal systems (Acha et al., 2004; Campagna et al., 2007; Belkin et al., 2009; Romero et al., 2006; Miloslavich et al., 2011; Alemany et al., 2009).

The San Jorge Gulf (SJG) is the largest semi-open basin located in Patagonian Shelf (approximately 40000 $km^2$), the most productive hydrocarbon (oil and gas) basin of Argentina and represents an important hydrocarbon reserves (Sylwan, 2001). It is located between 45° S and 47° S with depths slightly over 100 m in the central region (Fig. 1). Its broad mouth extends 230 km from Bahía San Gregorio to Cabo Tres Puntas along the meridian 65° 45' W, connecting the Gulf with the Argentine Continental Shelf through a sill that increases in the S-N direction, reaching a maximum depth of $\sim$ 60 m near 46° 48' S (Fig. 1). This geomorphological feature in interaction with the strong tidal currents increase near-bottom mixing that reaches the sea surface and produces changes of well-stratified conditions to well-mixed conditions within a few kilometers during the warm seasons.

The SJG circulation is driven by intense westerly winds and high amplitude tides (Palma et al., 2004; Tonini et al., 2006; Moreira et al., 2011). Estimates of tidal energy dissipation by bottom friction derived from numerical models results (Glorioso and Flather, 1995, 1997; Palma et al., 2004; Moreira et al., 2011) suggest that most of the dissipation occurs at the mouth of the SJG, mainly in the southeast region. The dissipation rate is high enough to break up the seasonal thermocline and give rise to the formation of an intense tidal front.

Due to its configuration and variability (Carbajal et al., 2018), the tidal front enhances the biological productivity nearby (Glembocki et al., 2015), plays a key role in the development of ecological processes and is closely related to fishery resources (Acha et al., 2015; Alemany et al., 2014). Studying the frontal variability, both spatial and temporal, is essential to understand the mechanism responsible for that enhancement and to define main frontal properties related to biological effects. Thus, the use of a high-resolution sampler system was key to evidence the high-frequency frontal variability. Knowledge of mesoscale variability is not only crucial to interpreting the biological influence of the fronts (Landeira et al., 2014), but it will also contribute to the establishment of new conservation strategies and the management of marine resources.

In this article we describe the cruise's design and the procedures used for the acquisition (Sect. 2), calibration (Sect. 3 and Sect. 4) and processing (Sect. 5) of the data set obtained during MARES leg 2. Furthermore, data gridding procedures of one high-resolution section collected during the frontal survey are presented in Appendix A.

**2  Field measurements and equipment**

Two types of surveys were carried out between Feb. 4-10 2014 on board of the Canadian research vessel (R/V) *Coriolis II* during MARES leg 2 cruise: one located in the STF region (Sect. 2.1) and another in a fixed position near the center of the Gulf mouth (Sect. 2.2). Complementary observations were also accomplished (Sect. 2.3). While towed undulating vehicle systems have been used by investigators (Twardowski et al., 2005; Brown et al., 1996), the PROMESse program was the ideal framework for the application of new technologies in Patagonian Shelf such as this vehicle, achieving unprecedented high temporal and spatial resolution data in the region, particularly in the STF. After the *Coriolis II* cruise, two surveys were carried out in the SJG (during Nov.2016 and Oct-Nov.2017) in the framework of a national project in which the towed undulating vehicle was not available. Thereby, the focus of this article is on the originality of the data set to study the high-frequency frontal dynamics.

Table 1 summarizes the characteristics of the sensors used in each instrument. Date and time from data sets are reported in Coordinated Universal Time (UTC).

**2.1 Southern Tidal Front observations**

Eighteen cross-front transects (six in late spring tide (Feb. 5), six in intermediate tide (Feb. 8-9) and six in early neap tide (Feb. 9-10) were occupied in the STF using a towed undulating vehicle EIVA Marine Survey Solutions model Scanfish II (http://aquaticcommons.org/3106/1/ACT_WR07-01_Tower_Vehicles.pdf), fitted with a modular CTD Sea-Bird Electronic (SBE) model 49FastCAT (16 Hz) and a combined fluorometer and turbidity sensor WetLabs model ECO FLNTU (8 Hz). The modular CTD has no memory nor internal batteries, and does not support auxiliary sensor inputs either. Therefore, the ECO FLNTU could not be directly associated with the CTD data during the surveys and thus the fluorescence and turbidity signal was acquired without the corresponding date, time and position. Nevertheless, it was possible to link both signals to get the missing data in a post-processing using complete data (date, time and position) of Scanfish CTD and the sampling intervals of the two sensors employed simultaneously to link the missing data to the corresponding record of the FLNTU data. The sections length ranged between 17.4 km to 63.1 km, which is equivalent to 1:09 h and 4:33 h of transit (see Table 2 for details). The sections occupied in late spring and early neap tide covered an area of approximately 29.8 km (NW-SE) by 15.1 km (NE-SW) during a semi-diurnal tidal cycle each (Figs. 1c and 1e, respectively), while the intermediate tide survey consisted of a single transect (T1) occupied six times, back and forth, also during a semi-diurnal tidal cycle (Fig. 1d). Surveys detail above are shown in Fig. 1. It is worth pointing out that the sampling experience shows that carrying out the same cross-front transect several times is recommendable (e.g. Fig. 1d) given that the along-front variability (ruled by baroclinic instabilities) difficult the prediction of the front´s location during the early neap tide and that the cross-frontal physical mechanisms (ruled by the tide) of interest do not change much at this scales. The horizontal separation of Scanfish sawtooth profiles was approximately 81 m–291 m, the latter largely dependent on bottom depth and the condition in the sea surface, descending (ascending) the vehicle at an absolute rate of nearly 0.9 m s$^{-1}$. On board, the towed vehicle was monitored through the roll and pitch sensors. The vehicle attitude was governed through two rear-mounted flaps and depths were provided by the CTD pressure sensor (Brown et al., 1996). The data collected with the Scanfish II provided a quasi-synoptic spatial and temporal resolution to characterize the influence of the high/low tide behavior and determine the front displacements relative to the phase of the tide.

An additional cross-frontal transect was occupied across the STF (on Feb. 5 at night and Feb. 8 at late afternoon) to study the biological and chemical characteristics of the water column. Each realization consisted of five quasi-continuous full depth CTD-rosette casts spaced at distance intervals of $\sim$ 4.9 km, using a CTD SBE model SBE911*plus*, equipped with oxygen, pH, fluorescence, nutrients, photosynthetically active radiation (PAR), beam transmission and altimeter sensors (Table 1). The altimeter sensor was used to determine distance to the bottom. Most vertical profiles reached to within $\sim$ 9 m off the bottom. Oxygen, pH and fluorescence sensors were calibrated based on water samples from Niskin bottles as described in Sect. 3. Data from the remaining sensors are reported based on factory calibrations only. Down profiles are reported in this data set because during downcast the CTD sensors measure the water column with minimal interference from the underwater package.

**2.2 Fix Station observations**

From Feb. 6 2014 17:04 h UTC to Feb. 8 2014 04:01 h UTC, a time series was carried out in a fixed station (FS) near to the center of the SJG mouth (45° 56' S, 65° 33' W). The time series consisted of thirteen quasi-continuous full depth CTD-rosette casts collected approximately every 2:55 h during 34:57 h using the CTD SBE911*plus*. The objective of this survey was to monitor the pycnocline displacements in the water column and to determine the mechanisms responsible for these vertical movements. It is expected that these mechanisms could be similar to that observed in the STF and will allow to identify vertical stratification variations in the water column which could affect the productivity in the frontal region.

**2.3 Complementary observations**

An underway CTD SBE model SBE19*plus* (4 Hz) coupled with a Seapoint fluorescence sensor was used to identify the position and orientation of the STF and remained operational throughout the entire cruise (Sect. 6). The real-time sample interval of the underway was set to 0.25 s along the tracks. Direct velocity measurements were collected with a Teledyne RD Instruments (TRDI) 150 kHz Ocean Surveyor hull-mounted Acoustic Doppler Current Profiler (ADCP, Sect. 7). These data sets are reported in the data collection.

**3 Water Samples**

[revised manuscript text omitted]

*Acknowledgements.* This program was financed by the Institut des sciences de la mer de Rimouski/Université du Québec à Rimouski (IS-MER/UQAR) from Canada, the Consejo Nacional de Investigaciones Científicas y Técnicas (CONICET) and the Ministerio de Ciencia y Tecnología (MINCyT) from Argentina. We thank the Instituto Nacional de Investigación y Desarrollo Pesquero (INIDEP, Argentina) that made available the Autosal salinometer and the Laboratorio de Oceanografía Química y Contaminación de Aguas (LOQyCA, Argentina) for allowing the use of the water samples measurements of nutrients, pH and DO. We also acknowledge Valérie Massé-Beaulne's detailed explanation of the measurement of Chl concentrations. We thank the crew and scientific staff of R/V *Coriolis II*. Finally, we wish to acknowledge D. Valla for his useful comments and language editing.

**Table 1.** Summary of the sensors used on board R/V *Coriolis II* in MARES leg 2 (Feb. 4-10 2014).

| Instrument | Sensor | Model | Serial # | Calibration date |
|---|---|---|---|---|
| SBE911*plus* (CTD profile) | Temperature | SBE3*plus* | 5769 | Oct.22 2013 |
| | Pressure | Digiquartz with TC | 1168 | Nov.19 2013 |
| | Conductivity | SBE4 | 4244 | Nov.06 2013 |
| | Oxygen | SBE43 | 2766 | Nov.15 2013 |
| | pH | SBE18 | 1078 | Nov.20 2013 |
| | Fluorescence | ECO FL WetLabs | FLRT-3363 | Nov.11 2013 |
| | Nutrients | Satlantic MBARI-ISUS | 0184 | May.25 2013 |
| | PAR | Biospherical/Licor | 70455 | Nov.04 2013 |
| | Beam transmission | WetLabs C-Star | CST-1628PR | Jun.11 2013 |
| | Altimeter | PSA-916 | 61114 | - |
| SBE49FastCAT (Scanfish II) | Temperature | | 0226 | Jan.17 2011 |
| | Pressure | Strain gauge | 0226 | Jan.14 2011 |
| | Conductivity | | 0226 | Jan.17 2011 |
| | Fluorescence/Turbidity | ECO FLNTU WetLabs | FLNTURT-2037 | Oct.07 2010 |
| SBE19*plus* (Underway) | Temperature | | 4975 | Mar.07 2013 |
| | Pressure | Strain gauge | 4975 | Feb.28 2013 |
| | Conductivity | | 4975 | Mar.01 2013 |
| | Fluorescence | Seapoint Chlorophyll Fluorometer | 2803 | Apr.28 2006 |

**Table 2.** Field measurements across the STF using the towed undulating vehicle Scanfish II. Arrows indicate the cruise path parallel to each transect (⇒: offshore the Gulf, ⇐: into the Gulf). $\Delta x$ represents the transect length.

| | Transect | Path | Begin | | End | | Depth range (m) | $\Delta x$ (km) | # Scans |
|---|---|---|---|---|---|---|---|---|---|
| | | | Date (hh:mm) | lat./lon. (°) | Date (hh:mm) | lat./lon. (°) | | | |
| late spring tide | T1 | ⇒ | Feb.5 (07:38) | -46.512,-65.970 | (10:01) | -46.614,-65.709 | 0.83-79.87 | 24.0 | 114505 |
| | T2 | ⇐ | (10:13) | -46.592,-65.697 | (12:10) | -46.462,-66.015 | 2.40-81.36 | 28.8 | 111801 |
| | T3 | ⇒ | (12:21) | -46.441,-65.996 | (13:55) | -46.538,-65.742 | 2.73-79.95 | 22.4 | 90001 |
| | T4 | ⇐ | (14:08) | -46.512,-65.734 | (15:48) | -46.418,-65.961 | 6.11-79.96 | 21.2 | 96002 |
| | T5 | ⇒ | (15:59) | -46.401,-65.936 | (17:43) | -46.513,-65.653 | 6.81-80.24 | 25.4 | 99601 |
| | T6 | ⇐ | (17:59) | -46.488,-65.636 | (19:23) | -46.398,-65.862 | 4.17-79.74 | 20.4 | 81201 |
| intermediate tide | T1-1 | ⇒ | Feb.8 (23:58) | -46.502,-65.997 | Feb.9 (01:32) | -46.598,-65.755 | 0.50-82.97 | 21.8 | 90003 |
| | T1-2 | ⇐ | (01:46) | -46.596,-65.759 | (02:55) | -46.518,-65.953 | 1.54-84.37 | 17.4 | 66001 |
| | T1-3 | ⇒ | (03:05) | -46.518,-65.967 | (04:37) | -46.612,-65.720 | 1.30-82.71 | 22.1 | 88001 |
| | T1-4 | ⇐ | (04:45) | -46.614,-65.710 | (06:46) | -46.496,-66.012 | 2.66-80.68 | 27.3 | 116001 |
| | T1-5 | ⇒ | (07:07) | -46.499,-66.004 | (08:45) | -46.612,-65.718 | 2.74-80.44 | 25.5 | 94002 |
| | T1-6 | ⇐ | (09:00) | -46.615,-65.700 | (11:19) | -46.485,-66.036 | 2.04-85.13 | 30.4 | 134002 |
| early neap tide | T1 | ⇒ | (11:34) | -46.484,-66.036 | (13:48) | -46.615,-65.709 | 4.36-80.24 | 29.7 | 128801 |
| | T2 | ⇐ | (14:02) | -46.589,-65.696 | (16:00) | -46.462,-66.013 | 2.37-80.38 | 28.6 | 114001 |
| | T3 | ⇒ | (16:13) | -46.440,-65.997 | (18:06) | -46.563,-65.681 | 2.65-80.34 | 28.3 | 108801 |
| | T4 | ⇐ | (18:20) | -46.542,-65.659 | (21:02) | -46.394,-66.028 | 2.49-81.49 | 34.0 | 156001 |
| | T5 | ⇒ | (21:17) | -46.373,-66.009 | (23:26) | -46.513,-65.652 | 0.06-80.27 | 32.0 | 124002 |
| | T6 | ⇐ | (23:40) | -46.492,-65.633 | Feb.10 (04:13) | -46.225,-66.334 | 1.71-83.93 | 63.1 | 262406 |

[Figure]

**Figure 1.** Study area (a) Patagonia shelf; (b) San Jorge Gulf, the underway track for MARES leg 2 (red line), the Scanfish transects across the STF (black lines) and the FS (black triangle). A zoom of the survey across the STF for (c) late spring tide, (d) intermediate tide, showing the CTD vertical profiles (orange circles); and (e) for early neap tide. Bathymetry is shown as shaded colors, highlighting the bank south of the Gulf, where depths range from 45 to 75 m. Arrows in light gray indicate the cruise path, particularly in d) the double arrow references the survey back and forth over T1. STF: Southern Tidal Front. FS: Fixed Station.

[Figure]

**Figure 2.** Distribution of dissolved oxygen residuals versus dissolved oxygen concentration (both in $\mathrm{ml\,l^{-1}}$), before (orange dots) and after (blue dots) the SBE43 sensor calibration.

[Figure]

**Figure 3.** Relative frequency of (a) temperature, (b) conductivity and (c) fluorescence residuals before (gray shaded bars) and after (black solid bars) the underway calibrations.

[Figure]

**Figure A1.** Vertical section of temperature in late spring tide for transect 1. The horizontal scale represents distance in km from an arbitrary zero position, whereas the profile of the seabed is derived from the ship echo-sounder EK60. The consecutive 'V-shaped' profiles of the Scansifh II are marked in gray, with a dot every 10 data points, and the cruise path with a black arrow.